# Targeting *Echinococcus multilocularis* PIM kinase for improving anti-parasitic chemotherapy

**Akito Koike[1], Frank Becker[2], Peter Sennhenn[3], Jason Kim[4], Jenny Zhang[4], Stefan Hannus[2], Klaus Brehm[1]***

**1** University of Würzburg, Institute of Hygiene and Microbiology, Consultant Laboratory for Echinococcosis, Würzburg, Germany, **2** Intana Bioscience GmbH, Martinsried, Germany, **3** transMedChem Consulting, München, Germany, **4** Immuneering Corporation, Cambridge, Massachusetts, United States of America

\* kbrehm@hygiene.uni-wuerzburg.de

## Abstract

### Background

The potentially lethal zoonosis alveolar echinococcosis (AE) is caused by the metacestode larval stage of the tapeworm *Echinococcus multilocularis*. Current AE treatment options are limited and rely on surgery as well as on chemotherapy involving benzimidazoles (BZ). BZ treatment, however, is mostly parasitostatic only, must be given for prolonged time periods, and is associated with adverse side effects. Novel treatment options are thus urgently needed.

### Methodology/principal findings

By applying a broad range of kinase inhibitors to *E. multilocularis* stem cell cultures we identified the proto-oncogene PIM kinase as a promising target for anti-AE chemotherapy. The gene encoding the respective *E. multilocularis* ortholog, EmPim, was characterized and *in situ* hybridization assays indicated its expression in parasite stem cells. By yeast two-hybrid assays we demonstrate interaction of EmPim with *E. multilocularis* CDC25, indicating an involvement of EmPim in parasite cell cycle regulation. Small molecule compounds SGI-1776 and CX-6258, originally found to effectively inhibit human PIM kinases, exhibited detrimental effects on *in vitro* cultured parasite metacestode vesicles and prevented the formation of mature vesicles from parasite stem cell cultures. To improve compound specificity for EmPim, we applied a high throughput *in silico* modelling approach, leading to the identification of compound Z196138710. When applied to *in vitro* cultured metacestode vesicles and parasite cell cultures, Z196138710 proved equally detrimental as SGI-1776 and CX-6258 but displayed significantly reduced toxicity towards human HEK293T and HepG2 cells.

### Conclusions/significance

Repurposing of kinase inhibitors initially designed to affect mammalian kinases for helminth disease treatment is often hampered by adverse side effects of respective compounds on human cells. Here we demonstrate the utility of high throughput *in silico* approaches to

**Data Availability Statement:** The sequences of all genes newly characterized in this work have been submitted to the GenBank database and are available under the accession numbers ON005010

(empim) and ON005011 (emcdc25). Accession numbers of all other proteins or genes analysed in this work are listed in S3 Table. All other relevant data are in the manuscript and Supporting Information files.

**Funding:** This work was supported by the Wellcome Trust (https://wellcome.ac.uk/), grant [107475/Z/15/Z] (FUGI, to KB), and by a grant of the Bayerische Forschungsstiftung (https://www.forschungsstiftung.de/) [AZ-1341-18] (to SH, KB, and PS). AK was supported by a fellowship of the Heiwa Nakajima foundation (https://www.isc.tokushima-u.ac.jp/english/scholarships/3807/) under grant number [AK-1-2019]. The funders had no role in study design, data collection and analysis, decision to publish, or preparation of the manuscript.

**Competing interests:** The authors declare that they have no competing interests.

design small molecule compounds of higher specificity for parasite cells. We propose EmPim as a promising target for respective approaches towards AE treatment.

## Author summary

The larva of the tapeworm *E. multilocularis* grows tumor-like within the host liver, thus causing the lethal disease alveolar echinococcosis (AE). Anti-parasitic treatment relies on chemotherapy with benzimidazoles, which do not kill the parasite and must be applied for years. As druggable enzymes with key functions in growth control, protein kinases are promising drug targets and many kinase inhibitors have been identified during cancer research. Optimized for binding to human kinases, however, repurposing of such drugs for parasitic disease treatment is associated with adverse side effects. Herein, the authors applied an *in silico* approach to identify small molecule compounds that show higher specificity for a parasite kinase, EmPim, over its mammalian homologs. The authors demonstrate expression of EmPim in *Echinococcus* stem cells, which are the drivers of parasite growth, and show that mammalian PIM kinase inhibitors SGI-1776 and CX-6258 also affect parasite development *in vitro*. Finally, they show that one of the *in silico* screened compounds is equally effective against the parasite as SGI-1776 and CX-6258, but significantly less toxic to human cells. These results demonstrate the utility of *in silico* approaches to identify parasite-specific kinase inhibitors.

## Introduction

The metacestode (MC) larval stage of the cestode *E. multilocularis* is the causative agent of alveolar echinococcosis (AE), a potentially lethal zoonosis prevalent in the Northern Hemisphere [1]. Intermediate hosts (rodents, humans) are infected by oral uptake of infectious eggs, which contain an embryonic larval stage, called the oncosphere. Within the intestine of the intermediate host, the oncosphere hatches from the egg, penetrates the intestinal barrier, and gains access to the inner organs. Usually within the host liver, the oncosphere then undergoes a metamorphosis-like transition towards the MC stage [2]. The *E. multilocularis* MC consists of numerous vesicles, which grow infiltratively, like a malignant tumour, into the surrounding liver tissue, eventually resulting in organ failure if no adequate treatment is applied [3]. As we have previously shown, MC growth and proliferation strictly depend on a population of pluripotent stem cells (called 'germinative cells' (GC) in the case of *Echinococcus*), which, as typical for flatworms, are the only mitotically active cells of the MC and give rise to all differentiated cells [4]. Currently, the only option to cure AE is surgical removal of the invading MC tissue, supported by chemotherapy using benzimidazoles (BZ; albendazole, mebendazole), which target parasite β-tubulin [5]. Surgical removal of parasite tissue is, however, only possible in around 20% of cases, leaving BZ chemotherapy as the only remaining treatment option for non-operable patients [3]. Although the prognosis of such patients has significantly improved after the introduction of BZ chemotherapy around 40 years ago, adverse side effects are frequently observed [6]. Furthermore, albendazole and mebendazole appear to be effective against the parasite in only one third of cases [7], whereas they are parasitostatic only in the majority of patients and must, therefore, be applied for years to decades, sometimes even lifelong [8]. Since GC are the only mitotically active cells of the MC [4], it has already been proposed that the high recurrence rates after anti-AE chemotherapy are due to limited activity of

BZ against the parasite's stem cell population [9]. Hence, alternative drugs are urgently needed, which are also active against the *Echinococcus* GC [5].

Due to their catalytical process in transferring phosphate onto protein targets, protein kinases (PKs) are particularly druggable enzymes [10] and most small molecule compounds that interfere with kinase activity bind to the ATP binding pocket [11]. Furthermore, PKs are crucially involved in regulating proliferation and differentiation of eukaryotic cells, making them highly attractive targets for strategies to chemically interfere with aberrant cell proliferation, e.g. in the context of malignant transformation [12]. One of these PKs, the PIM (proviral integration site for murine leukemia virus) kinase, has recently drawn much attention as a potential target for treating multiple forms of cancer [13–20]. Mammals typically express three PIM kinase isoforms, Pim-1, Pim-2, and Pim-3, which are constitutively active serine/threonine kinases [21] that phosphorylate numerous protein substrates and are downstream effectors of a variety of cytokine signalling pathways [22]. Via activation of CDC25 phosphatase, mammalian Pim-1 is involved in the regulation of the cell cycle [23,24] and aberrant expression of PIM kinases has been associated with numerous forms of malignant transformation [22]. A hallmark of PIM kinases is their unusual hinge region, which facilitates the development of specific kinase inhibitors and, during recent years, several small molecule compounds with activities against PIM kinases have been identified. Of these, the imidazole pyridazine compound SGI-1776 proved to be an effective pan-PIM inhibitor with $IC_{50}$ values of 7, 363, and 69 nM against human Pim-1, Pim-2, and Pim-3, respectively, but also inhibited the kinases FLT3 (44 nM) and haspin (34 nM) [16]. Among second generation inhibitors, the oxindole-based compound CX-6258 displayed even higher specificity for PIM kinases than SGI-1776 ($IC_{50}$ Pim-1, 5 nM; Pim-2, 25 nM, Pim-3, 16 nM) [25] and less activity against FLT3 ($IC_{50}$: 134 nM) [26]. At least in melanoma cell lines, CX-6258 also showed activities against haspin kinase, although also at much higher $IC_{50}$ values than against Pim-1 and Pim-3 [27]. Although SGI-1776 has proceeded to clinical phase I trials against non-Hodgkin lymphoma and prostate cancer, the respective studies have been terminated due to toxicity in cardiac electric cycle prolongation [28], probably due to their activities against PIM kinases in non-cancer cells and/or to off-target effects.

In the present work, we demonstrate that SGI-1776 and CX-6258 also exert detrimental effects on *in vitro* cultivated stem cells and larval stages of *E. multilocularis*. We characterized the *Echinococcus* PIM kinase ortholog, show that it contains the majority of amino acid residues that mediate the binding of PIM inhibitors to the ATP binding pocket, and demonstrate that, like the mammalian counterpart Pim-1, the *Echinococcus* enzyme interacts with CDC25 phosphatase. Using an *in silico* modelling approach to discriminate between mammalian and parasite PIM sequences, we then identified compound Z196138710, which displays equal toxicity against parasite larvae as SGI-1776 or CX-6258, but which is much less toxic for mammalian cells. The impact of these findings on drug design strategies against AE are discussed.

## Methods

### Ethics statement

*In vivo* propagation of parasite material was performed in mongolian jirds (*Meriones unguiculatus*), which were raised and housed at the local animal facility of the Institute of Hygiene and Microbiology, University of Würzburg. This study was performed in strict accordance with German (*Deutsches Tierschutzgesetz*, *TierSchG*, version from Dec-9-2010) and European (European directive 2010/63/EU) regulations on the protection of animals. The protocol was approved by the Ethics Committee of the Government of Lower Franconia (Regierung von Unterfranken) under permit numbers 55.2–2531.01-61/13 and 55.2.2-2532-2-1479-8.

## Organisms and culture methods

All experiments were performed with the *E. multilocularis* isolates H95 and GH09 [29] which either derive from a naturally infected fox of the region of the Swabian Mountains, Germany (H95) [30] or from Old World Monkey species (*Macaca fascicularis*) that had been naturally infected in a breeding enclosure (GH09) [31]. The isolates were continuously passaged in mongolian jirds (*Meriones unguiculatus*) essentially as previously described [32,33]. *In vitro* culture of parasite metacestode vesicles (MV) under axenic conditions was performed as previously described [34] and the isolation and maintenance of *Echinococcus* primary cell cultures (PC) was carried out essentially as established by Spiliotis et al. [35]. In all cases, media were changed every three to four days (d), including addition of fresh compound solution in the case of inhibitor studies. For these studies, specific concentrations of compounds, dissolved as 10–50 mM stock solutions and stored at -80°C, were added to parasite cultures as indicated, and as negative control DMSO (0.1%) was used. SGI-1776 was purchased from Selleckchem (Houston, Texas) and CX-6258 was from Cayman chemical (Ann Arbor, Michigan). Z196138710 was purchased from SIA Enamine (Riga, Latvia). Providers of other kinase inhibitors are listed in S1 Table. A6 medium was prepared by seeding $1.0 \times 10^6$ rat Reuber hepatoma cells [33] in 175 cm$^2$ culture flasks with 50 ml DMEM (Dulbecco's Modified Eagle Medium) + GlutaMAX-I (life technologies) including 10% Fetal Bovine Serum Superior (life technologies) and incubated for 6 d under aerobic condition. The supernatant was sterile filtrated to remove hepatocytes. Similarly, B4 medium was prepared by seeding $1.0 \times 10^7$ rat Reuber hepatoma cells in 175 cm$^2$ culture flasks with 50 ml DMEM+GlutaMAX-I including 10% FBS and incubated for 4 d under aerobic conditions prior to sterile filtration.

## Anti-parasitic inhibitor assays

In cell viability assays for initial screening of kinase inhibitors, PC were isolated from mature MV using a previously established protocol [34] and PC density was measured by densitometry. 1 Unit (U) of PC is defined as the amount which increases OD$_{600}$ by 0.01.15 U of isolated PC (~$2.25 \times 10^3$ cells/well) were seeded into 384 well plates with 100 µl of conditioned medium (50% A6 medium + 50% B4 medium) including defined concentration of inhibitors as indicated. Plates were incubated at 37°C under nitrogen atmosphere. After 3 d, cell viability was measured using the Cell Titer Glo 2.0 cell viability assay (Promega), according to the manufacturer's instructions. According to Crouch et al. [36], Kangas et al. [37], and Maehara et al. [38], the strength of the luminescent signal in this test is directly proportional to the amount of ATP and to the number of viable cells in culture. Luminescence was measured using a Spectramax iD3 Multi-mode Microplate reader (Molecular Devices; San Josè, CA, USA). Measured luminescence units were normalized to those of the DMSO control and visualized as heatmaps with GraphPad Prism version 9.3.1 (Graphpad software). All kinase inhibitors were tested independently in three technical replicates.

 In mature MV assays, 10 individual MV each were cultured in 2 ml of conditioned medium (100% A6 medium) in the presence of inhibitors in 12 well plates under axenic conditions for 28 d as described in [39]. Only vesicles without visible brood capsules or protoscoleces were used, all vesicles had comparative diameter (~0.5 mm) and were of comparative age (5–7 months of culture). Structural integrity of MV was assessed using an optical microscope (Nikon eclipse Ts2-FL). Experimental set-up and execution of inhibitor studies and structural integrity assessment was performed by independent experimenters. Criteria for intact or damaged vesicles were essentially as previously described [40,41]. All experiments were performed using 3 biological replicates. Percentages of structurally intact vesicles were statistically

analyzed with one-way ANOVA with Dunnet's multiple comparison tests in Graphpad prism 9.3.1(Graphpad software).

In vesicle formation assays, PC were isolated as described above and 100 U of isolated PC (~$1.5 \times 10^4$ cells/well) were seeded in 96 well plates with 200 μl of conditioned medium (50% A6 medium + 50% B4 medium) for 21 d under nitrogen atmosphere. The number of newly formed vesicles was counted using an optical microscope (Nikon eclipse Ts2-FL) essentially as previously described [41]. Kruskal-Wallis test followed by Dunn's multiple comparisons test was used in GraphPad Prism version 9.3.1(Graphpad software) to analyze the difference of vesicle numbers in control and treatment groups. In this analysis, all concentrations were compared with the negative control DMSO. Experiments with SGI-1776 and CX-6258 were performed in three biological and technical replicates. Experiments with SGI-1776 and Z196138710 were performed in three technical replicates.

## Mammalian cell cultivation and drug treatment

The toxicity of inhibitors against mammalian cells was evaluated by treatment of the commonly used cell lines HEK293T [42] and HepG2 [43], which were cultivated and passaged as described in [43,44]. Semi-confluent cultured cells up to ten passages after vial thawing were trypsinized and $1.0 \times 10^3$ cells were seeded in 384 well plates with 50 μl of DMEM (Dulbecco's Modified Eagle Medium) + GlutaMAX-I (life technologies) including 10% Fetal Bovine Serum Superior (life technologies). 24 h after seeding, 50 μl of DMEM+-GlutaMAX-I including FBS and inhibitors were added. Final concentrations of inhibitors were adjusted to 0–30 μM as indicated, DMSO alone was used as a control. Plates were incubated for 3 d under aerobic conditions and cell viability was measured using the Cell Titer Glo 2.0 system (Promega) according to the manufacturer's instructions. Luminescence was measured by a Spectramax iD3 Multi-mode Microplate reader (Molecular Devices). Three independent experiments with three technical replicates were carried out for both cell lines. Luminescence units were normalized to the DMSO control of each independent experiment and expressed as percentage of luminescence unit. One-Way-ANOVA test followed by Tukey's multiple comparisons test was used in GraphPad Prism version 9.3.1 (Graphpad software) for statistical analysis. Based on the cell viability data, dose-response curves were drawn and $IC_{50}$ (best fit value) were calculated using GraphPad Prism version 9.3.1 (Graphpad software).

## Nucleic acid isolation, cloning and sequencing

RNA isolation from *in vitro* cultivated axenic MV and PC was performed using a Trizol (5Prime, Hamburg, Germany)-based method as previously described [45]. For reverse transcription, 2 μg total RNA was used for cDNA synthesis using oligonucleotide CD3-RT (5'-ATC TCT TGA AAG GAT CCT GCA GGT$_{26}$ V$^{-3'}$). PCR products were cloned using the PCR cloning Kit (QIAGEN, Hilden, Germany) or the TOPO XL cloning Kit (Invitrogen). The complete list of primer sequences used for *empim* and *emcdc25* cDNA amplification and characterization is given in S2 Table. Upon cloning, PCR products were directly sequenced using primers binding to vector sequences adjacent to the multiple cloning site by Sanger Sequencing (Microsynth Seqlab, Göttingen, Germany). The sequences of all genes newly characterized in this work have been submitted to the GenBank database and are available under the accession numbers ON005010 (*empim*) and ON005011 (*emcdc25*). Accession numbers of all other proteins or genes analysed in this work are listed in S3 Table.

### *In situ* hybridization and 5-ethynyl-2'-deoxyuridine (EdU) labeling

Digoxygenin (DIG)-labeled probes were synthesized by *in vitro* transcription with T7 and SP6 polymerase (New England Biolabs), using the DIG RNA labelling kit (Roche) according to the manufacturer's instructions from *empim*-cDNA fragments cloned into vector pJET1.2 (Thermo Fisher Scientific). Primers for probe production are listed in S2 Table. Probes were subsequently purified using the RNEasy Mini Kit (Qiagen), analysed by electrophoresis, and quantified by dot blot serial dilutions with DIG-labeled control RNA (Roche). Whole-mount *in situ* hybridization (WISH) was subsequently carried out on *in vitro* cultivated metacestode vesicles as previously described [4], using vesicles (isolate H95) of at least 1 cm in diameter to avoid losing material during washing steps. Fluorescent specimens were imaged using a Nikon Eclipse Ti2E confocal microscope and maximum projections created using ImageJ as previously described [46]. In all cases, negative control sense probes yielded no staining results. *In vitro* labelling with 50 μM EdU was done for 5, 8, or 16 hours (h) and fluorescent detection with Alexa Fluor 555 azide was performed after WISH essentially as previously described [4]. Series of pictures were taken at randomly chosen sections of the germinal layer of 5 MC vesicles with 40 × objective lens as Z-stack. Among the picture of each Z-stack, the layer of strongest signal was selected by the function of Z project in Fiji/ImageJ and processed [47]. EdU positive cells, WISH positive cells and double positive cells were counted manually and independently. The number of cells with each signal were calculated to cell number per $mm^2$ on the germinal layer.

### Yeast two hybrid (Y2H) analyses

The Gal4-based Matchmaker System (Takara Bio, USA) was used as described by Zavala-Góngora et al. [48,49] and Stoll et al [46]. Full-length cDNAs encoding EmPim kinase and EmCdc25 phosphatase were PCR amplified from parasite cDNA using primers as listed in S2 Table and cloned into pGADT7 or pGBKT7 (Takara/Clontech). The *Saccharomyces cerevisiae* Gold strain (Clontech) was transformed with these plasmids by a one-step protocol described in Tripp et al. [50] and inoculated on Leu⁻/Trp⁻ double dropout agarose plates. After incubation at 30˚C for 2 d, three colonies were picked from each transformation and inoculated independently in 2 ml of liquid Leu⁻/Trp⁻ medium and incubated at 30˚C and 200 rpm until above $OD_{660}$ = 1.0. Yeast cultures were then diluted to equalized densities of $OD_{660}$ = 1.0, 0.1 and 0.01. Diluted yeast cultures were then dropped (5 μl each) onto Leu⁻/Trp⁻/His⁻ triple dropout plates and Leu⁻/Trp⁻/Ade⁻/His⁻ quadruple dropout plates. After 48–72 h incubation at 30˚C, pictures of plates were taken with ProtoCOL SR colony counter (Synbiosis). The pictures were then converted into gray scale and processed using Fiji/imageJ [47] according to the protocol described in [51]. The level of growth on quadruple dropout plates with the inoculation density $OD_{660}$ = 1.0 was quantified as gray value. The quantified level of growth was statistically analysed with one-way ANOVA with Tukey's multiple comparison tests in Graphpad prism 9.3.1 (Graphpad software).

### Bioinformatic procedures

Amino acid sequences of human Pim-1, Pim-2, Pim-3, Cdc25A, Cdc25B, and Cdc25C were used as queries in BLASTP searches against the *E. multilocularis* genome on WormBase ParaSite [29,52,53]. Reciprocal BLASTP searches were performed again the SWISSPROT database as available under the KEGG database at Genomenet [54]. Domain structure was analyzed with SMART8.0 [55–57]. Percent identity/percent similarity values of amino acid sequences were calculated through Sequence Manipulation Suite [58]. Multiple sequence alignments were performed using Clustal omega [59] or CLUSTALW2.1 [60] in MEGA11 [61] under the

following settings: Gap Opening Penalty = 10.00, Gap Extension Penalty = 0.20, Delay Divergent Cutoff = 30%. Aligned sequences were visualized by SnapGene Viewer (Snapgene software). Based on these alignments, phylogenic trees were generated by MEGA11. The statistical method for tree building was maximum likelihood, substitution model was Jones-Taylor-Thompson model, ML Heuristic method was Nearest-Neighbor Interchange. All transcripts of *Echinococcus* genes were analysed using Integrative Genomics Viewer [62,63] and previously published transcriptome data [29] to check for correct prediction of the sequences available through UniProt.

For virtual compound identification procedures, the target of interest (EmPim) was screened against Fluency, a proprietary deep learning-based platform [64]. The EmPim sequence tested was retrieved from Uniprot, functional domains were retrieved from the Pfam database. The target of interest was screened against 2 small molecule libraries: the Enamine Hinge Binders library [65] (n = 24,000), and Enamine Diverse REAL drug-like library [66] version 2021q1-2, further filtered for drug-like properties based on Lipinski's rule of 5 [67] (n = 21.4M). We applied predictions from 2 versions of the Fluency model, termed model 1 and model 2, which were trained on varying data sets and settings. Every combination of protein, compound library, and model was predicted by Fluency, resulting in 4 sets of predictions, ranking molecules from strongest predicted binder to weakest. Out of the 200 top-ranked *in silico* hits from the Fluency screen, 20 compounds were selected for purchase and profiling based on their Fluency Screen score, diversity of structures representing best the chemical space of the 200 hits and molecular modeling in hPIM (6VRU) employing SeeSAR modeling software from BioSolveIT (version 11.2.). The generated poses were assessed for a meaningful binding mode into the ATP-pocket, absence of intra- and intermolecular clashes and torsion quality.

Modeling of SGI-1776 and Z196138710 against human Pim-1 was performed using modeling software SeeSAR Version 12.0.1 (BioSolveIT). To create broad diversity and to continue further, 200 poses were demanded within SeeSAR [68] per structure; the integrated Analyzer Mode was used to select those poses that were clash-free and only exhibited green torsions, i.e., torsions that are statistically prevalent in small-molecule crystal structures [69].

## Results and discussion

### Initial screen of broad-range kinase inhibitors against *E. multilocularis* cell cultures

Based on sequence similarities between the catalytic domains and the presence of accessory domains, conventional PKs are basically divided into seven sub-groups, against which specific kinase inhibitors are available [70]. Homologs belonging to all these sub-groups are also expressed by *E. multilocularis* [29]. To identify kinase sub-groups that are important for *Echinococcus* stem cell function, we carried out an initial screen of 14 available kinase inhibitors covering all sub-groups, against *E. multilocularis* PC, which are strongly enriched in germinative (stem) cells [4] (Fig 1). To assess for direct cytotoxic effects, we performed cell viability assays with all inhibitors at a concentration of 10 μM and measured cell viability after 3 d and 7 d of drug exposure. As shown in Fig 1, several inhibitors against the AGC and the CAMK groups showed clear effects against *Echinococcus* PC, whereas Dasatinib, directed against the BCR-Abl subfamily of tyrosine kinases, even stimulated parasite cell proliferation under these conditions. The strongest anti-parasitic effect was achieved using the PIM-specific inhibitor SGI-1776, with approximately 60% and 80% growth inhibition after 3 d and 7 d, respectively. In all further experiments we therefore decided to concentrate on the PIM kinase family and their role in *Echinococcus* stem cell biology.

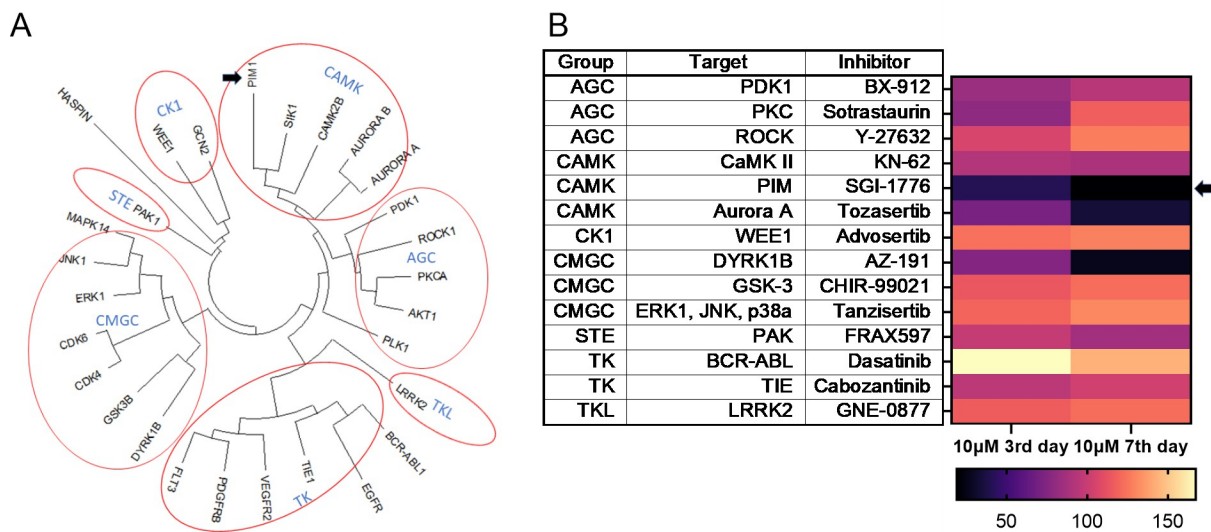

**Fig 1. Activities of selected kinase inhibitors against *E. multilocularis* PC.** (A) Phylogenetic tree of human PKs, based on homologies within the kinase domain. Seven groups according to current nomenclature [71] are indicated. (B) Heatmap showing the effects of different kinase inhibitors, covering all 7 groups, on *E. multilocularis* PC. Colour-code below indicates percentage of luminescence signal (i.e. number of viable cells), normalized to signals from DMSO controls, after 3 d and 7 d of incubation with 10 µM of inhibitor. Inhibitor names, human target proteins, and kinase sub-families are indicated in the table to the left. Black arrow indicates the pan-PIM kinase inhibitor SGI-1776.

## Characterization of an *E. multilocularis* PIM kinase

Since SGI-1776 was originally designed to inhibit human PIM kinases and showed strong effects against *Echinococcus* PC, we were interested in characterizing parasite PIM kinase orthologs. To this end, we performed BLASTP analyses using all three human PIM isoforms as queries against the published *E. multilocularis* genome sequence [29]. In all three cases we identified one single locus (EmuJ_000197100), which encoded a protein with significant homologies. Reciprocal BLASTP analyses against the SWISSPROT database using the EmuJ_000197100 gene product as a query revealed high homologies to human Pim-1, particularly within the kinase domain (47% identical, 65% similar residues) (Fig 2). Significant homologies were also detected between the kinase domains of the EmuJ_000197100 gene product and PRK2 (38%, 58%) and PSK2 (30%, 50%), which are PIM kinase orthologs of *Caenorhabditis elegans* and yeast, respectively. We thus named the *Echinococcus* gene *empim*, encoding the serine/threonine kinase EmPim (657 amino acids, 74 kDa theoretical MW). Since BLASTP searches against the *E. multilocularis* genome using the amino acid sequences of EmPim or all three human PIM kinase isoforms did not reveal any other gene with significant homologies, we concluded that *empim* is a single copy gene and that, in contrast to mammals, *E. multilocularis* only encodes one single PIM kinase isoform.

We then conducted similar analyses for the genome of the related trematode parasite *Schistosoma mansoni*, and, likewise, found one single locus (Smp_090890) encoding a PIM ortholog with significant homologies to EmPim, which we named SmPim (S1 Fig). The presence in the genome of just one gene encoding a PIM kinase ortholog appears to be a specific trait for parasitic flatworms since, as already mentioned, mammals express three PIM kinase isoforms [72,73]. Furthermore, three isoforms have already been described to be expressed by the related, but free-living, planarians, in which they belong to the group of 'immediate early genes', the expression of which is drastically upregulated during wound-induced responses

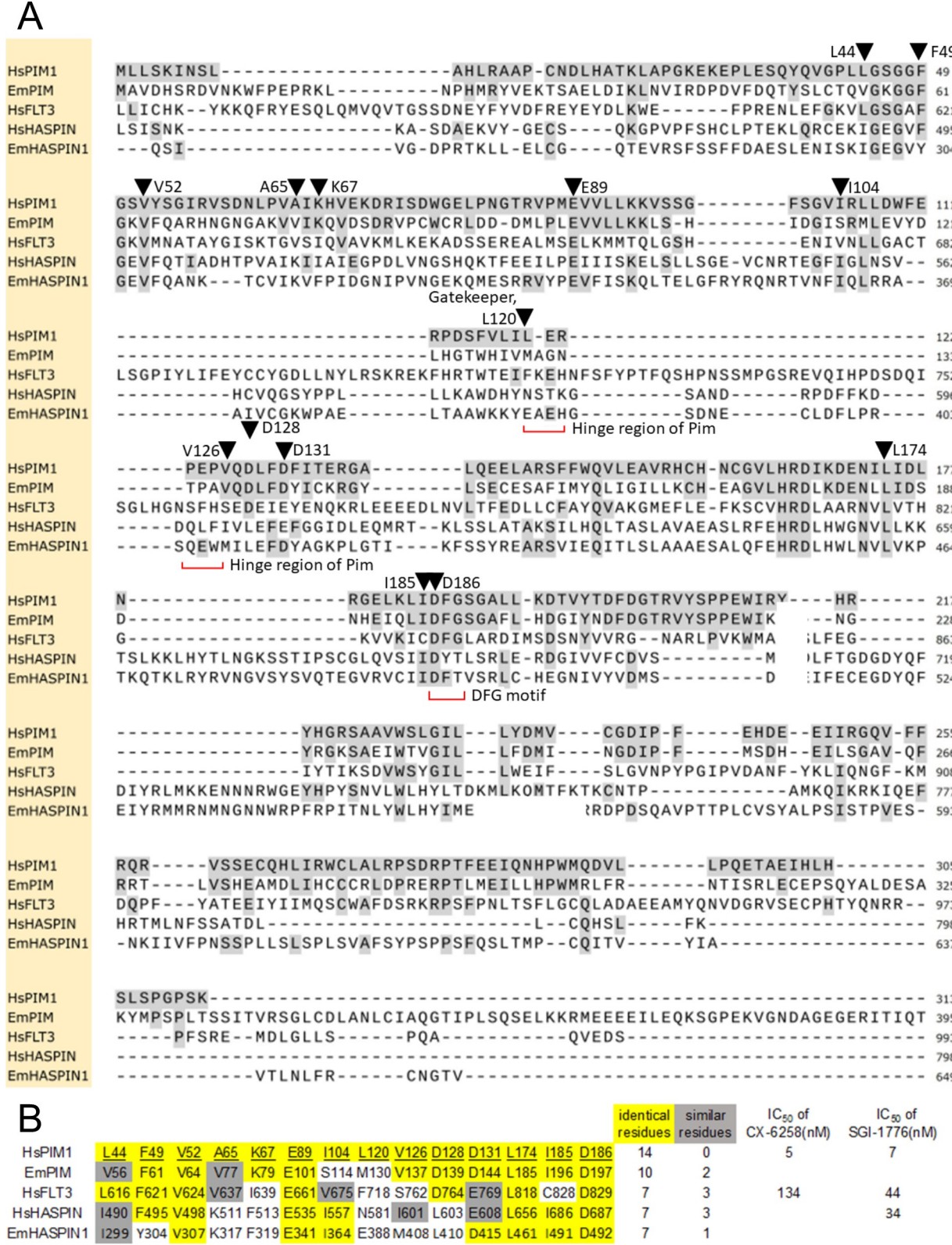

**Fig 2. Homologies and structural features of EmPim.** (A) Amino acid sequence alignment of the kinase domains of human Pim-1 (HsPIM1), *E. multilocularis* Pim (EmPim), human FLT3 kinase (HsFLT3), human haspin kinase (HsHASPIN) and an *E. multilocularis* haspin kinase ortholog (EmHASPIN1). Residues identical to human Pim-1 are shown in black on grey. Kinase DFG motifs and the hinge regions are marked in red. Black triangles indicate residues known to be involved in the interaction between human Pim-1 and compound CX-6258 (numbered

according to human Pim-1). (B) Presence of amino acid residues important for the interaction between human Pim-1 and CX-6258 in different kinases. For each of the 14 known residues of human Pim-1 (HsPIM1), the corresponding residue and position in *E. multilocularis* Pim (EmPIM), human FLT3 kinase (HsFLT3), human haspin kinase (HsHASPIN), and the *E. multilocularis* haspin kinase isoform (EmHASPIN1) are shown. Residues identical to those of human Pim-1 are marked in yellow, residues with similar biochemical properties are marked in green. The numbers of identical/similar residues compared to human Pim-1 are listed to the right as well as $IC_{50}$ values of compounds CX-6258 and SGI-1776 to human enzymes.

[74]. Most interestingly, both EmPim and SmPim not only harbour the typical serine/threonine kinase domain but also a large C-terminal extension (S1 Fig), which is missing in human and planarian PIM kinase isoforms. Due to the absence of a regulatory domain, the human PIM kinase isoforms are considered constitutively active kinases, the activity of which is largely regulated at transcriptional, translation, and proteosomal degradation level [21,75]. Hence, in contrast to these enzymes, the PIM kinases of parasitic flatworms are likely subject to more elaborate regulatory mechanisms, although we could not identify consensus regulatory regions, such as conserved phosphorylation sites, within the C-terminal extension.

Although the precise mode of interaction between SGI-1776 and PIM kinases is not known, crystallographic studies have already been conducted for the binding mode of CX-6258 to human Pim-1 [26]. These studies identified 14 amino acid residues of particular importance for the inhibitor-kinase interaction (indicated in Fig 2). Interestingly, of these 14 residues, 10 are invariantly present in EmPim and two further residues represent exchanges with conserved biochemical properties (Fig 2). In the case of human FLT3 kinase, which is also inhibited to a certain extent by CX-6258, only 7 of these residues are conserved (Fig 2). We thus propose that CX-6258 (and most probably also SGI-1776) binds to EmPim with intermediate affinities when compared to Pim-1 and FLT3. Notably, an FLT3 ortholog is apparently not expressed by *E. multilocularis* since BLASTP genome mining using mammalian FLT3 as a query did not reveal clear orthologs. This is supported by phylogenetic studies indicating that FLT3 kinases have evolved before the chordate/urochordate split, but after the divergence of protostomes and deuterostomes [76].

Apart from the tyrosine kinase FLT3, both SGI-17776 [16] and CX-6258 [27] have been demonstrated to inhibit, although with lower affinities, the kinase haspin (haploid germ cell-specific nuclear protein kinase), which mediates histone modification in mammals [77]. Interestingly, a haspin ortholog is also expressed by the *E. multilocularis* genome (EmuJ_000667600). Within the kinase domain, both the *Echinococcus* and the human *haspin* kinases share 6 or 7, respectively, of the 14 residues involved in the CX-6258-kinase interaction (Fig 2). It thus cannot be excluded that the parasite haspin kinase might also be targeted by CX-6258, at least to a certain extent.

Taken together, our analyses indicated that *E. multilocularis* expresses a single PIM ortholog with substantial homologies to mammalian PIM kinases within the kinase domain. Unlike PIM kinases of mammals or planarians, the PIM kinase enzymes of parasitic flatworms contain a large C-terminal extension, indicating a more complex mode of regulation than in the case of conventional (mammalian) PIM kinases. Based on the conservation of the majority of amino acid residues that mediate binding of CX-6258 to PIM kinases, we also concluded that available PIM kinase inhibitors should primarily target EmPim in *Echinococcus* cells. We cannot rule out, however, that part of the effects of CX-6258 (and of SGI-1776) described below might be due to inhibition of the *Echinococcus* haspin ortholog.

## Expression of *empim* in *Echinococcus* stem cells

In mammalian cells, PIM kinases are involved in cell proliferation and cell cycle regulation [78]. Since the germinative (stem) cell population is the only mitotically active *Echinococcus* cell type [4], we would expect expression of *empim* in germinative cells if EmPim has

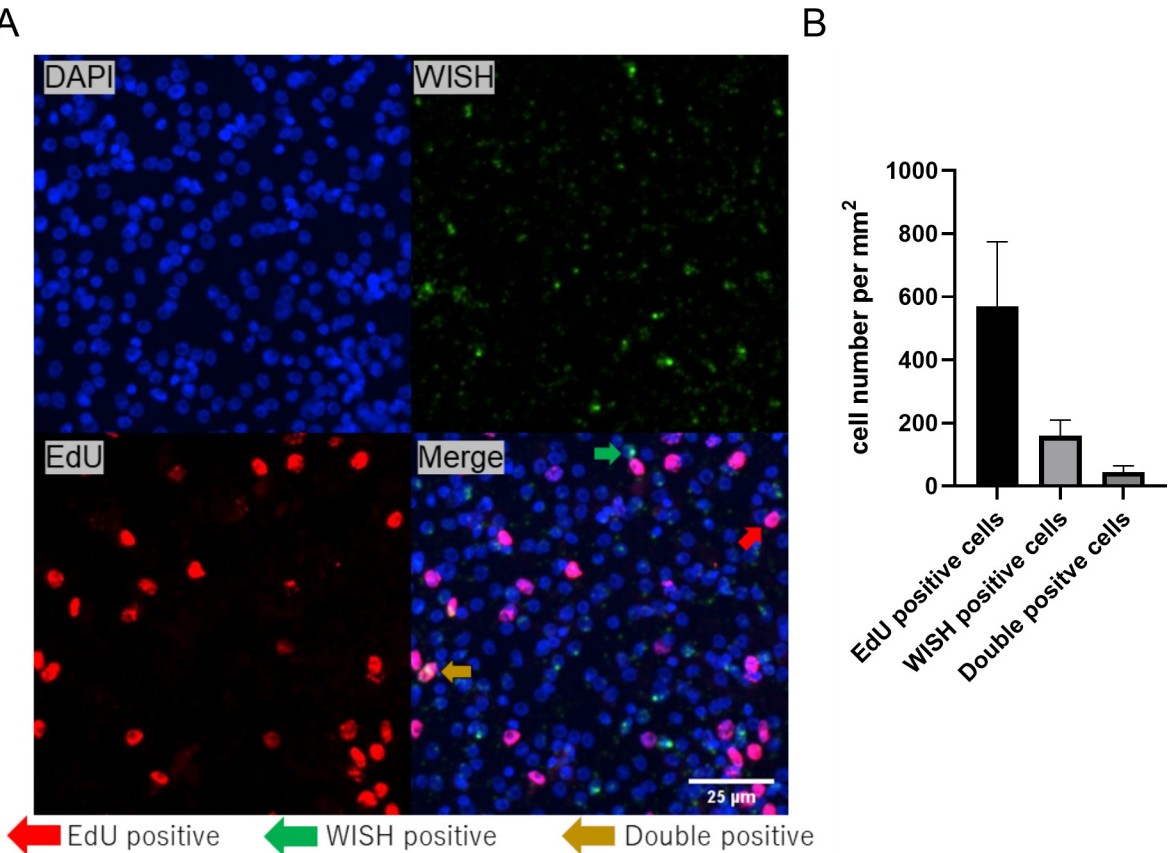

**Fig 3. Expression of *empim* in *Echinococcus* MV.** (A) WISH on *E. multilocularis* MV directed against *empim*. Channels shown are DAPI (blue, nuclear staining), WISH (green, *empim*+), EdU (red, S-phase stem cells), and merge as indicated. Green arrow indicates example of *empim*+/EdU- cell, red arrow indicates example of *empim*-/EdU+ cell, yellow arrow indicates example of *empim*+/EdU+ cell. Size bar represents 25 μm. A schematic illustration of MV regions where images have been taken is given in S3 Fig. (B) Average numbers of *empim* +/EdU+ cells per mm$^2$ of germinal layer are shown. Error bar indicates standard deviation.

equivalent functions as mammalian PIM kinases. According to transcriptome analyses that had been produced during the *E. multilocularis* whole genome project [29], *empim* displayed higher expression in primary cell cultures after 2 d of incubation when compared to metacestode vesicles (S2 Fig). Since these cultures are highly enriched in germinative cells [4], we assumed that *empim* might show a dominant expression in this cell type. To clarify the situation, we carried out WISH analyses on MC vesicles that had been incubated with EdU, thus identifying the proliferating stem cell compartment. As shown in Fig 3, in the germinative layer of *in vitro* cultivated MV we detected *empim* signals in both EdU+ and EdU- cells. After an 8 h EdU pulse, around 25% of *empim*$^+$ cells were also EdU$^+$. For the majority of *empim*+ cells, however, we could not detect co-staining with EdU, indicating that they either represent post-mitotic, differentiated cells, or stem cells which were not in S-phase during the EdU pulse.

Previous studies on human chronic myelogenic leukemia cells indicated that Pim-1 is cell cycle-regulated with highest expression levels at G1-S and G2-M transitions, whereas a significant drop in Pim-1 expression occurs during S-phase [79]. Since EdU exclusively stains cells that have been in S-phase during the 8 h pulse, it is thus possible that the fraction of germinative cells which express *empim* is significantly higher than 25%, provided that the PIM kinase gene is also cell cycle-regulated in *Echinococcus*. In any case, our WISH/EdU experiments clearly indicate that a certain fraction of parasite stem cells expresses *empim*.

## Interaction between EmPim and CDC25C phosphatase

Modulation of mammalian cell cycle progression through Pim-1 is mainly mediated by phosphorylation, and thereby activation, of dual specific phosphatases of the CDC25 family [23,24]. CDC25 phosphatases are highly conserved from yeast to mammals, are expressed in differing numbers of isoforms (e.g. 3 in humans, 1 in yeast, 2 in *Drosophila*, 4 in *C. elegans*), and induce the M-phase of the cell cycle by removing inhibitory phosphates from cyclin-dependent kinases [23,80,81]. Provided that EmPim, despite its unusual C-terminal extension, also mediates cell cycle progression in *Echinococcus*, we would expect that it physically interacts with CDC25 isoforms in this parasite. To investigate these aspects, we first mined the available *E. multilocularis* genome sequence for the presence of CDC25 encoding genes. Using either of the three human CDC25 isoforms (CDC25A-C) as a query against the *E. multilocularis* genome in BLASTP analyses, we constantly identified one single locus (EmuJ_001174300) encoding a 762 amino acid (theoretical MW = 85,3 kDa) protein tyrosine phosphatase. The EmuJ_001174300 product displayed relatively weak overall homologies to CDC25 family members of humans, insects, or nematodes, but contained a Rhodanese domain, which is a hallmark of CDC25 family M-phase inducers [82] (Fig 4). Furthermore, amino acid residues within the Rhodanese domain that are critical for enzymatic function were highly conserved between the EmuJ_001174300 product and human CDC25 orthologs (Fig 4). Furthermore, in reciprocal BLASTP analyses against the SWISSPROT database using the EmuJ_001174300 gene product as a query, we detected highest homologies with CDC25 orthologs of mammals and invertebrate model organisms. We thus concluded that *E. multilocularis* genome contains a single locus encoding a CDC25 family phosphatase and named the respective gene *emcdc25* (encoding the protein EmCDC25).

To investigate possible interactions between EmPim and EmCDC25 we employed the yeast two-hybrid (Y2H) system which we had previously used to study protein-protein interactions between *Echinococcus* factors [46,48,49,83]. To this end, we cloned the full-length cDNAs for EmPim and EmCDC25 into vectors pGBKT7 and pGADT7, respectively, and assessed colony growth under medium (triple dropout plates) and high (quadruple dropout plates) stringency conditions. As shown in Fig 5, under medium stringency conditions we obtained growth for the combination EmPim-pGBKT7 x EmCDC25-pGADT7 but we also observed some growth capacity for EmPim-pGBKT7 with the empty vector control. Under high stringency conditions, on the other hand, only EmPim-pGBKT7 x EmCDC25-pGADT7 yielded positive results, indicating specific interaction between these proteins. Statistically significant differences between EmPim-pGBKT7 x EmCDC25-pGADT7 and empty vector controls were also observed in quantitative assays measuring yeast growth on quadruple dropout plates (OD660 = 1.0) (Fig 5). We thus concluded that, like in mammalian systems, the *Echinococcus* Pim kinase acts upstream of a CDC25 family phosphatase. Whether this interaction is involved in *Echinococcus* M-phase entry control remains to be established. Due to the expression of EmPim in *Echinococcus* stem cells and the high conservation of Pim/CDC25-dependent M-phase entry control from yeast to mammals [23,24], such a role is, however, highly likely.

We cannot yet tell whether the EmPim-EmCDC25 interaction indeed involves phosphorylation of the M-phase regulator by the PIM kinase, although this is clearly the case for human Pim-1 and CDC25A [23]. Currently available PIM kinase activity assays rely on small peptide substrates basing on known target consensus sequences (K/R-K/R-R-K/R-L-S/T-a; a = small amino acid residue) for human PIM kinases [84]. Unfortunately, we could not identify a sequence motif in EmCDC25 that exactly matches the consensus of human PIM kinases, thus making it very difficult to establish a functional EmPim kinase assay at present. Hence, further investigations are necessary to clearly define phosphorylation sites for EmPim to facilitate kinase assays e.g. for high throughput screening.

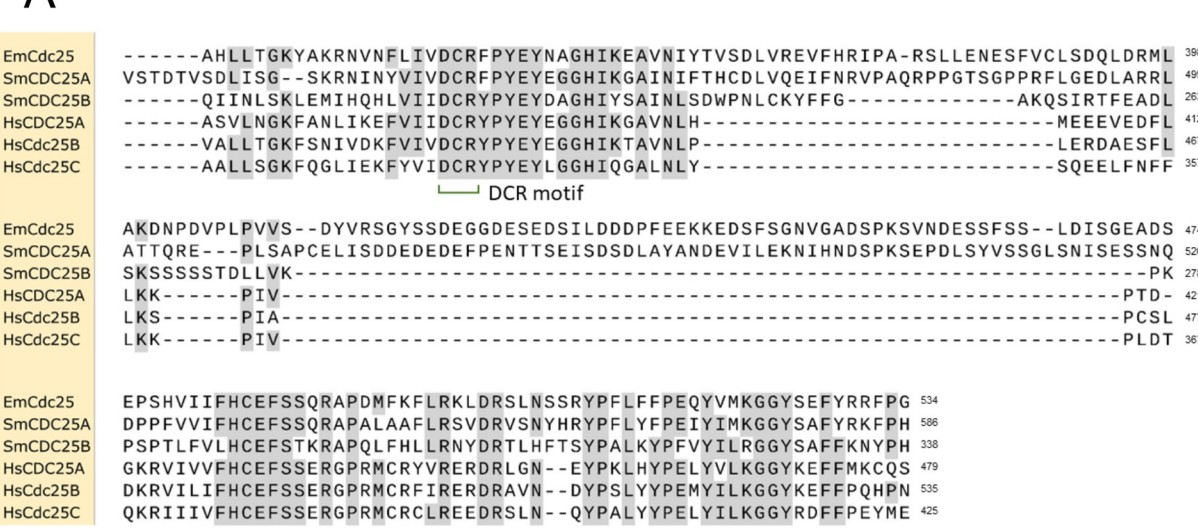

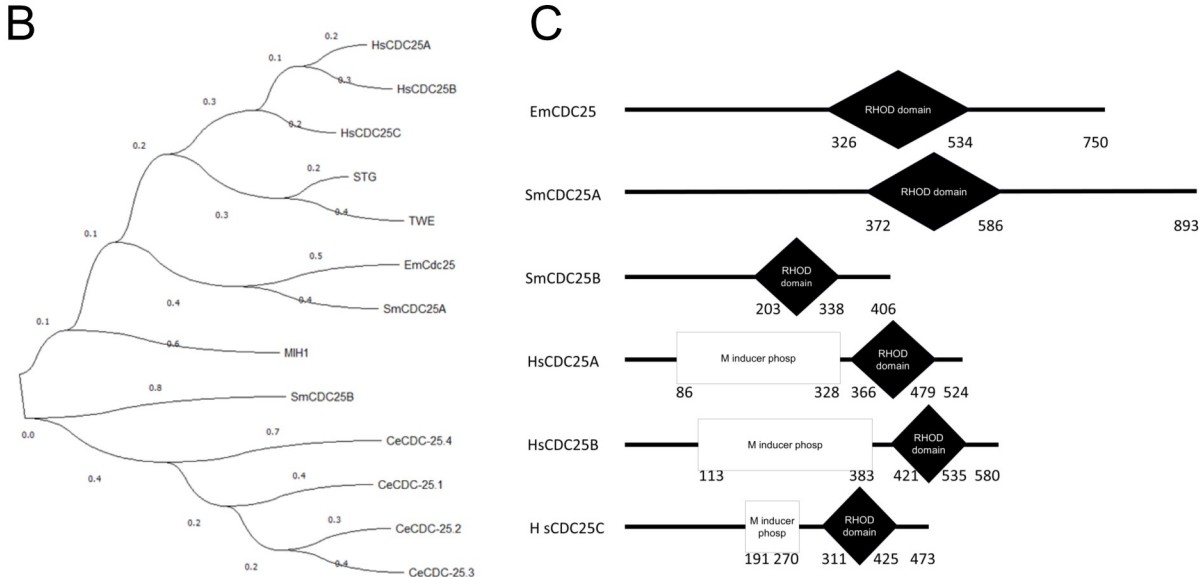

**Fig 4. Domain structure and homologies of EmCDC25.** (A) Amino acid sequence alignment of the Rhodanese homology domains of EmCDC25 (EmCdc25), two *S. mansoni* CDC25 orthologs (SmCDC25A, SmCDC25B), and three human CDC25 orthologs (HsCDC25A-C). Conserved Rhodanese domain DCR motifs and the active site are indicated. Residues identical to EmCDC25 are shown in black on grey. (B) Phylogenetic tree based on Rhodanese domains of different CDC25-like phosphatases. Sequences derived from *E. multilocularis* (EmCDC25), *S. mansoni* (SmCDC25A/B), *H. sapiens* (HsCDC25A-C), *C. elegans* (CeCDC25 1–4), *D. melanogaster* (TEW, STG), and *Saccharomyces cerevisiae* (MIH1). Statistical method for the tree was maximum likelihood (ML), substitution model was Jones-Taylor-Thompson, ML heuristic method was Nearest Neighbour Interchange. (C) Domain structures of EmCDC25, two different CDC25 orthologs of S.mansoni (SmCDC25A/B), and three human CDC25 isoforms (HsCDC25A-C). Shown are Rhodanese domains and M-phase inducer phosphatase domains, which are typical for mammalian isoforms.

## Effects of SGI-1776 and CX6258 on *Echinococcus* larvae and stem cells

Thus far, we had only measured effects of SGI-1776 on *Echinococcus* primary cells in a cell viability assay. However, the actual target for anti-AE therapy are MC vesicles. Furthermore, for effective elimination of parasite tissue, the capacity of stem cells to differentiate into MV

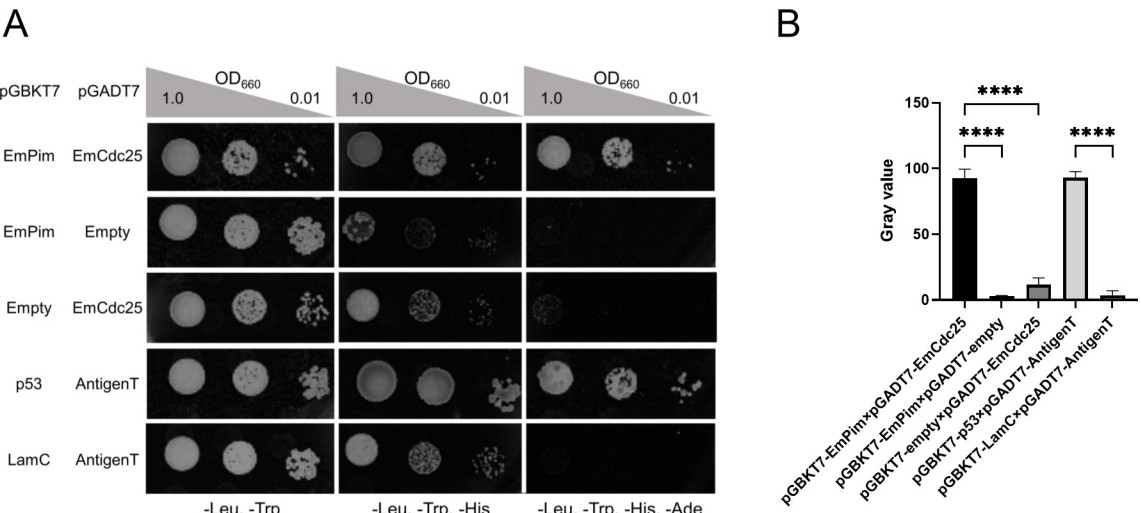

**Fig 5. Interaction between EmPim and EmCDC25.** (A) Representative pictures of yeast transformant growth on plates selecting for plasmids (-Leu, -Trp) as well as triple dropout (-Leu, -Trp, -His) and quadruple dropout (-Leu, -Trp,—His, -Ade) plates for interaction under medium and high stringency conditions, respectively. Plasmid combinations are indicated to the right, $OD_{600}$ values for dropout density above. (B) Quantitative assay measuring growth densities of yeast transformants. Plasmid combinations are indicated below the graph. Error bar represents standard deviation. Tukey's multiple comparison test, followed by one way ANOVA was used to compare all experimental combinations, but only comparisons to the corresponding control are shown. **** indicates $p \leq 0.0001$.

vesicles must be eliminated [5]. We thus employed in further experiments previously established *in vitro* cultivation systems for mature MV and for the production of MV from stem cells. Furthermore, since EmPim contained the majority of residues that mediate the interaction between Pim kinases and CX-6258, we also included this inhibitor in our analyses. As shown in Fig 6, both SGI-1776 and CX-6258 had a detrimental and dose-dependent impact on the structural integrity of mature MV. Although incubation of MV with 3 μM of both inhibitors for 28 d did not lead to statistically significant effects, a concentration of 10 μM of these inhibitors led to a drastic loss of structural integrity of all (CX-6258) or almost all (SGI-1776) vesicles (Fig 6). In the case of 3 μM of these inhibitors, many vesicles lost structural integrity but still had the germinative layer attached to the parasite surface laminated layer (Fig 6). In the case of 10 μM of both inhibitors, however, complete detachment of the parasite tissue from the laminated layer was observed (Fig 6).

Since even after loss of structural integrity, parasite vesicles can in theory still harbor living stem cells, we then tested both inhibitors for their capacity to affect the formation of MV from cultivated stem cells. As shown in Fig 6, both inhibitors affected MV formation from stem cells in a dose-dependent manner. In the case of SGI-1776, vesicle formation, which in this system is usually achieved after 21 d [33], was completely prevented in the presence of 30 μM SGI-1776, and almost completely in the presence of 10 μM. At 3 μM concentration, SGI-1776 did not lead to statistically significant effects. CX-6258, on the other hand, already drastically affected MV formation at 3 μM and completely inhibited MV development at higher concentrations (Fig 6).

Taken together, these analyses demonstrated clear detrimental effects of both PIM kinase inhibitors on *Echinococcus* larvae and stem cells. Based on the homologies of EmPim to human PIM kinases in regions that are important for inhibitor-kinase interaction (Fig 2), we concluded that most of these effects should be due to an inhibition of EmPim, although we cannot fully exclude that a certain degree of inhibition of the *Echinococcus haspin* kinase might have contributed.

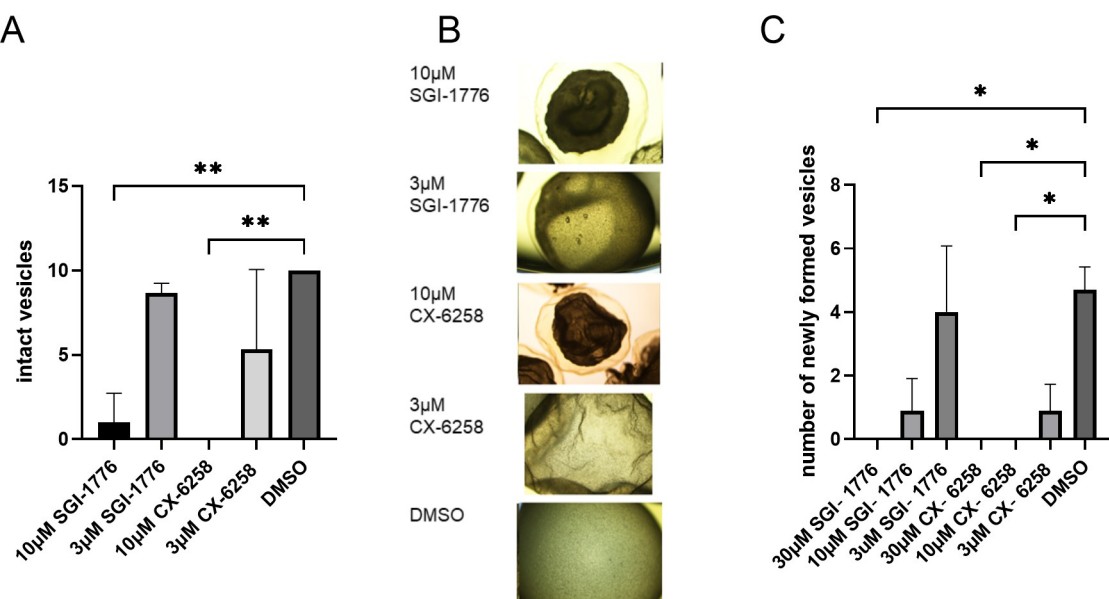

**Fig 6. Effects of SGI-1776 and CX-6258 on MV and PC.** (A) Inhibitor effects on mature MV. *E. multilocularis* MV were incubated for 28 d in the presence of different inhibitor concentrations as indicated below (with medium and inhibitor replacement every 3–4 d), and the number of structurally intact MV was inspected microscopically. One way ANOVA followed by Dunnet's multiple comparison test was used for comparison to the control DMSO group. ** indicates $p \leq 0.0021$. (B) Representative examples of MV incubated with different concentrations of inhibitors as indicated to the left. (C) Inhibitor effectos on the formation of MV from PC. Parasite stem cell cultures were incubated for 21 d in the presence of different inhibitor concentrations as indicated below. Numbers of fully mature MV were subsequently counted. Error bars represent standard deviation. Kruskal-Wallis test followed by Dunn's multiple comparison test was used for comparisons with control (DMSO) group. * represents $p \leq 0.0332$.

### *In silico* screening of EmPim inhibitors and effects against *Echinococcus* larvae

Due to their effects on human kinases, the utilization of currently available PIM inhibitors for chemotherapeutic approaches is associated with severe adverse effects [28,85–87]. At least in the case of SGI-1776, clinical trials against different forms of cancer had to be terminated since adverse effects on the cardiac electric cycle of patients were observed (NCT0084860, NCT01239108). We thus aimed at the identification of small molecule compounds that more specifically interact with the parasite enzyme isoforms when compared to human PIM kinases.

To this end, we first employed a very recently established *in silico* approach, the Fluency computational platform [64], which predicts quantitative binding affinities of compounds to target enzymes exclusively from amino acid sequences. Briefly, *Fluency* input consists of a protein amino acid sequence with domains optionally defined, and a small molecule structure in the form of SMILES. For each input-pair, Fluency predicts the protein-molecule binding affinity. Therefore, a natural application of Fluency is virtual screening of large molecular libraries against a target of interest to prioritize a top list of tractable size for downstream analysis such as medicinal chemistry analysis, docking, and experimental validation.

In a first Fluency screen of roughly 24 million compounds, using the EmPim amino acid sequence as a query, we obtained a list of 19,000 potential binders with predicted affinities between 10 nM and 1 μM for the parasite protein. Out of the 200 top-ranked *in silico* hits (S4 Table), 20 compounds were then selected for profiling based on (i) the Fluency screen score; (ii) diversity of chemical structures; and (iii) molecular modelling using the seeSAR software,

thus assessing the ATP pocket binding mode as well as the absence of intra- and intermolecular clashes (S5 Table).

We then tested the 20 selected compounds against *E. multilocularis* MV and stem cells. First, we again employed the MV assay and found 4 of the compounds (Z30898879, Z196138710, Z65225039, Z354576500) being highly effective in inducing structural vesicle damage at concentrations of 3 and 10 µM (Fig 7). These 4 compounds were then employed in

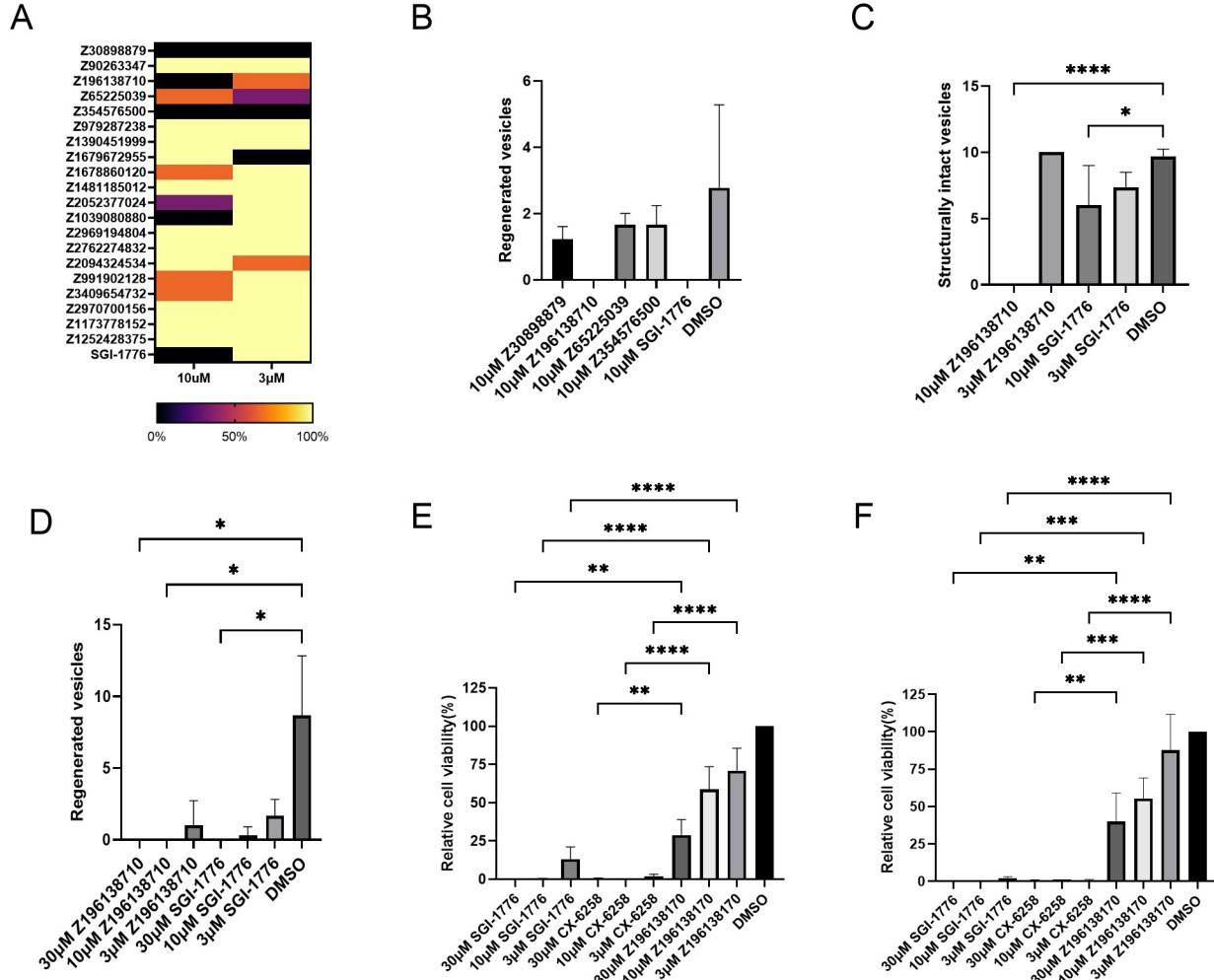

**Fig 7. Effects of Pim inhibitors on *Echinococcus* larvae and human cells.** (A) Heat map showing the effects of 20 *in silico* screen compounds on MV. Different concentrations (indicated below) of each compound (indicated to the left) were incubated in vitro with MV for 28 d and structural integrity was assessed. Colour-code indicating percentages of surviving vesicles is indicated below. (B) Effects of four *in silico* screen compounds on MV production from PC. 10 µM of each compound (indicated below) were incubated for 21 d with PC *in vitro* and the production of MV was assessed. For comparison, SGI-1776 was tested at 10 µM. Error bars represent standard deviation. Kruskal-Wallis test followed by Dunn's multiple comparison test was used for comparisons with the control (DMSO) group. (C) Effects of Z196138710 and SGI-1776 on MV. Both compounds were tested at different concentrations (shown below) on MV *in vitro*. Structural integrity was measured after 28 d. Error bar represents standard deviation. One was ANOVA followed by Dunnet's multiple comparison test was used in comparisons with control (DMSO) group. p values less than 0.0001 are summarized with **** and p values less than 0.0332 are summarized with *. (D) Effects of Z196138710 and SGI-1776 on the *in vitro* formation of MV from PC. Both inhibitors were incubated at different concentrations (indicated below) for 21 d with PC and the formation of mature MV was measured. Error bar represents standard deviation. Kruskal-Wallis test followed by Dunn's multiple comparison test was used in comparisons with control (DMSO) group. * represents p ≤ 0.0332. (E) Effects of Z196138710, SGI-1776, and CX-6258 on human HEK293T cells. HEK293T cells were incubated with different concentrations of inhibitors as indicated below. Cell viability was measured after 3 d. (F) Effects of inhibitors on human HepG2 cells. For experimental procedure, see (E). Error bar represents standard deviation. Tukey's multiple comparison test followed by one way ANOVA was used to compare all experimental settings, only comparisons for equal inhibitor concentrations are shown. P values less than 0.0001 are summarized with **** and p values less than 0.0021 are summarized with **.

the PC vesicle formation assay, leading to the identification of compound Z196138710 which, at a concentration of 10 μM, completely prevented MV formation (Fig 7). Finally, we focussed on the thienopyrimidine Z196138710 (*N*-(4-(difluoromethoxy)-3-methoxybenzyl)-thieno-[3,2-*d*]-pyrimidin-4-amine) and tested it in comparison to SGI-1776 on mature MV and the PC cultivation system for MV development. As shown in Fig 7, a concentration of 10 μM Z196138710 led to structural disintegration of 100% of MV after 28 d, which was even more effective than SGI-1776. In the case of MV development from primary cells, Z196138710 showed effects similar to those of SGI-1776 (Fig 7).

## Effects of Pim inhibitors on human cell lines

Having shown that Z196138710 shows similar (PC) or even higher (MV) toxicity towards *E. multilocularis* than SGI-1776, we were, in a final set of experiments, interested in possible toxicities of the thienopyrimidine compound on human cells. To this end, we employed the cell lines HEK293T and HepG2, which, according to previous studies, strongly depend on functional PIM kinases for cell viability [88–90]. As shown in Fig 7, at concentrations of 30 and 10 μM, both SGI-1776 and CX-6258 fully eliminated HEK293T and HepG2 cells within 3 d, whereas at the same concentrations, Z196138710 only inhibited both cell lines to 40–60% (Fig 7). At all three concentrations tested, Z196138710 showed weaker toxicity than SGI-1776 and CX-6258 against both cell lines, and these differences were statistically significant (Fig 7). This was also reflected in $IC_{50}$ analyses, in which Z196138710 yielded values of 11 and 16 μM against HEK293T and HepG2 cells, respectively, whereas $IC_{50}$ values in the case of SGI-1776 and CX-6258 were around 1 μM (S4 Fig). To assess whether the lower toxicity of Z196138710 towards human cell lines was due to reduced binding of the thienopyrmidine compound to human PIM kinases, we finally performed *in silico* modelling assays of Z196138710 and SGI-1776 on the structure of Pim-1. As shown in S5 Fig, these analyses revealed a binding affinity of SGI-1776 in the nanomolar range, which is in line with the results of previous biochemical assays [16]. For Z196138710, on the other hand, binding affinities in the micromolar range were obtained (S5 Fig), indicating that the low toxicity of the thienopyrimidine compound towards human cell lines is due to low binding to human PIM kinases. Although biochemical assays for measuring EmPim activity in the presence of kinase inhibitors will have to be established to verify these *in silico* analyses, our data at least point to Z196138710 as a promising candidate of an anti-*Echinococcus* compound with low adverse side effects.

In summary, we herein characterized an *E. multilocularis* single copy gene, which is expressed in parasite stem cells, and which encodes a PIM kinase family member that interacts with an *Echinococcus* CDC25 ortholog in Y2H assays. These data at least point to a role of EmPim in *Echinococcus* cell cycle regulation, which appears to be one of the conserved functions of PIM kinases in vertebrate and invertebrate organisms [23,24,91]. An important role of EmPim in *Echinococcus* stem cell function is further supported by our data on the detrimental effects of known PIM kinase inhibitors, SGI-1776 and CX-6258, on *in vitro* cultivated MV and, particularly, PC, which are highly enriched in stem cells [4]. Since EmPim shares 85% of amino acid residues that are critical for inhibitor binding to mammalian Pim-1, it is highly likely that these effects are primarily due to the inhibition of EmPim, although a certain level of off-target effects, which might involve a parasite *haspin* ortholog, cannot be fully excluded. Since the germinative (stem) cells are the crucial cell type for parasite growth within the host [4], molecules that regulate their proliferative capacity are, *per se*, attractive targets for anti-parasitic chemotherapy, provided that small molecule compounds can be identified which discriminate between these factors and their (usually) highly conserved mammalian orthologs. In the case of *Echinococcus*, high-throughput screening approaches towards the identification of

specific inhibitory compounds from extensive small-molecule libraries are hampered by the fact that the complex conditions of parasite cultivation, particularly those for stem cell cultures, only allow parallel screening of dozens to maybe a few hundreds of molecules, and usually must be carried out over several weeks. Even though elegant approaches such as the PGI-assay for measuring MV integrity [92] or PC-based cell activity assays [93] allow compound screening against *E. multilocularis* in shorter time, a pre-selection of molecules from complex compound libraries is still necessary to narrow down screening procedures to manageable sizes. We herein combined a novel, target-based computational approach and *in silico* modeling techniques to select 20 compounds from complex libraries of roughly 24 million molecules. Of these 20 compounds, 4 displayed detrimental effects on *in vitro* cultivated parasite larvae and stem cells, and one of these, Z196138710, even out-matched known inhibitors against the target kinase family concerning side effects on immortalized human cells. Our *in silico* approach thus effectively narrowed down the number of potential inhibitor molecules to a size that can be handled by *in vitro* approaches.

The true capacity of Z196138710 as chemotherapeutic agent against AE still has to be established in future studies. Those studies should include biochemical activity assays against Pim kinases and, of course, *in vivo* testing in murine models for echinococcosis [93]. The concentrations of Z196138710 which were effective against the parasite in our studies (between 3 μM in the case of stem cell cultures and 10 μM against mature MV) are well within the range of comparable molecules that were effectively used in murine *in vivo* assays. SGI-1776, for instance, displayed *in vitro* IC$_{50}$ values around 1 μM against HEK293T and HepG2 cells (this study) and 3 μM against a broad variety of other cell types [94], including chronic lymphocytic leukemia cells [16]. Although upon *in vivo* application in mice around 95% of the compound were bound to plasma proteins, it was still possible to achieve well tolerated plasma concentrations around 3 μM of free compound (not bound to plasma proteins), which were effective against solid tumour xenografts in mice [94,95]. The toxicity of SGI-1776 in clinical trials against non-Hodgkin lymphoma and prostate cancer was attributed to off-target effects on the human Ether-á-go-go-related (hERG) cardiac ion channel [95], a common problem in drug development strategies. Interestingly, in an *in silico* approach similar to the one used by us, Xu et al. [95] identified a hit compound with IC$_{50}$ of 52 μM against human Pim-1 which, however, did not bind to hERG. By combining the properties of the hit compound and SGI-1776, together with the introduction of systematic chemical modifications, these authors were then successful in developing compounds with very good affinity to Pim-1, but without the unwanted side effects on hERG [95]. Through the application of similar strategies, it should thus be feasible to develop Z196138710 into a chemical compound with potential against AE. Experiments towards this aim are currently undertaken in our laboratory.

## Supporting information

**S1 Table. Providers of small molecule compounds and inhibitors used in this study.**
(XLSX)

**S2 Table. Sequences of primers used in this study.**
(XLSX)

**S3 Table. Accession numbers of genes and proteins analysed in this study.**
(XLSX)

**S4 Table. Top 200 list of compounds after Fluency *in silico* screening against EmPim.**
(XLSX)

**S5 Table. Structures and features of 20 compounds selected after Fluency *in silico* screening.**
(XLSX)

**S1 Fig. Structural features and homologies of EmPim and SmPim.** EmPim. (A) Amino acid sequence alignment of the kinase domains of *E. multilocularis* Pim (EmPim), *S. mansoni* Pim (SmPim), and the three human Pim isoforms (HsPim1-3). Residues identical to human Pim-1 are shown in black on grey. Kinase DFG motifs and the hinge regions are marked in red. Black triangles indicate residues known to be involved in the interaction between human Pim-1 and compound CX-6258 (numbered according to human Pim-1). (B) Phylogenetic tree based on the kinase domains of EmPim, SmPim, all three human Pim kinases (HsPIM1-3), *C. elegans* PRK2, and yeast PSK2. (C) Domain composition and length of EmPim, SmPim, and human Pim kinases (HsPIM1-3). The total length of the proteins is shown to the right. The positions of the kinase domain are indicated.
(PDF)

**S2 Fig. Expression levels of *empim* in PC and MV.** Depicted are the expression values of *empim* according to Next Generation transcriptomic analyses performed by [29]. Values are given as transcripts per kilobase million (tpm). Shown are values for primary cells after 2 d cultivation and for MV without brood capsules (MV) as indicated. For comparison, values for the Polo like kinase encoding gene *emplk1* are shown, which is strictly expressed in *E. multilocularis* germinative stem cells [9]. Please note that for each condition only one sample has been analyzed (n = 1).
(PDF)

**S3 Fig. Schematic image of metacestode vesicles regions analyzed in *in situ* hybridization experiments.** Image refers to microscopi images shown in Fig 3. Brown circle (distal) indicates the acellular laminated layer, thick yellow circle (proximal indicates the germinative layer of a metacestode vesicle. Blue lines indicate picture plane for stack analysis (2 μm per stack) as indicated to the right. A series of images in the germinative layer was taken as Z stack by confocal microscopy, and the image of the strongest signal (highest cell density) was analyzed.
(PDF)

**S4 Fig. Dose response curves of PIM kinase inhibitors on human cell lines.** Cells were treated with 1 30 μM of SGI 1776 CX 6258 and Z 196138710 (as indicated) for 3 days and cell viability was measured. Signal intensities of each well were normalized to those of control samples treated with DMSO and shown as percentage. Error bar represents standard deviation. Shown are results for HEK 293 T (A) and HepG 2 (B) cells as mM concentration with $LogIC_{50}$ and $IC_{50}$ as indicated to the right One-Way- ANOVA test followed by Tukey's multiple comparisons test was used for statistical analysis.
(PDF)

**S5 Fig. SeeSAR analysis of SGI-1776 and Z196138710 binding to human Pim-1.** (A)
(PDF)

## Acknowledgments

The authors wish to thank Monika Bergmann and Dirk Radloff for excellent technical assistance.

## Author Contributions

**Conceptualization:** Akito Koike, Frank Becker, Peter Sennhenn, Jason Kim, Jenny Zhang, Stefan Hannus, Klaus Brehm.

**Data curation:** Akito Koike, Frank Becker, Peter Sennhenn, Jason Kim, Jenny Zhang, Stefan Hannus, Klaus Brehm.

**Formal analysis:** Akito Koike, Frank Becker, Peter Sennhenn, Jenny Zhang, Stefan Hannus, Klaus Brehm.

**Funding acquisition:** Frank Becker, Peter Sennhenn, Stefan Hannus, Klaus Brehm.

**Investigation:** Frank Becker, Peter Sennhenn, Jason Kim, Jenny Zhang, Stefan Hannus, Klaus Brehm.

**Methodology:** Akito Koike, Frank Becker, Peter Sennhenn, Jason Kim, Jenny Zhang, Stefan Hannus, Klaus Brehm.

**Project administration:** Frank Becker, Stefan Hannus, Klaus Brehm.

**Resources:** Stefan Hannus.

**Software:** Frank Becker, Peter Sennhenn, Jason Kim, Jenny Zhang.

**Supervision:** Stefan Hannus, Klaus Brehm.

**Validation:** Akito Koike, Frank Becker, Peter Sennhenn, Jason Kim, Jenny Zhang, Stefan Hannus, Klaus Brehm.

**Visualization:** Akito Koike, Frank Becker, Peter Sennhenn, Stefan Hannus, Klaus Brehm.

**Writing – original draft:** Akito Koike, Peter Sennhenn, Klaus Brehm.

**Writing – review & editing:** Stefan Hannus.

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
