## [Decision Letter · Decision Letter 0]

5 Jul 2022

Dear Prof. Brehm,

Thank you very much for submitting your manuscript "Targeting Echinococcus multilocularis PIM kinase for improving anti-parasitic chemotherapy" for consideration at PLOS Neglected Tropical Diseases. As with all papers reviewed by the journal, your manuscript was reviewed by members of the editorial board and by several independent reviewers. The reviewers appreciated the attention to an important topic. Based on the reviews, we are likely to accept this manuscript for publication, providing that you modify the manuscript according to the review recommendations. 

The reviewers felt that the manuscript was interesting, well-written and relevant to the field. They have provided useful suggestions in terms of writing more detailed descriptions of how the vesicles were prepared, generating an IC50 curve for Z196138710, and adding any potential data on in vivo toxicity. It would be preferred if additional experiments could be performed to test the in vivo toxicity, but as the reviewers indicate, please at least search and describe PK data, where available. 

Please also ensure that statistical tests are briefly described in each of the in the figure captions where relevant (Fig 3B, 5B, 6A, 6C, 7B-F). Although most of these are described in the text, 3B is does not seem to be, and the rest could just be identified with, for example, “(one way ANOVA, with Tukey’s multiple comparisons test)” after the P value descriptions. Additionally, reviewer 2 recommends separating the results and discussion. While that may improve clarity somewhat, it would be acceptable to submit these combined as currently written, and that editing decision will be left up to the authors.

Sincerely,

Bruce A. Rosa

Associate Editor

Makedonka Mitreva

Deputy Editor

The reviewers felt that the manuscript was interesting, well-written and relevant to the field. They have provided useful suggestions in terms of writing more detailed descriptions of how the vesicles were prepared, generating an IC50 curve for Z196138710, and adding any potential data on in vivo toxicity. It would be preferred if additional experiments could be performed to test the in vivo toxicity, but as the reviewers indicate, please at least search and describe PK data, where available. 

Please also ensure that statistical tests are briefly described in each of the in the figure captions where relevant (Fig 3B, 5B, 6A, 6C, 7B-F). Although most of these are described in the text, 3B is does not seem to be, and the rest could just be identified with, for example, “(one way ANOVA, with Tukey’s multiple comparisons test)” after the P value descriptions. Additionally, reviewer 2 recommends separating the results and discussion. While that may improve clarity somewhat, it would be acceptable to submit these combined as currently written, and that editing decision will be left up to the authors.

Reviewer's Responses to Questions

**Key Review Criteria Required for Acceptance?**

**Methods**

-Are the objectives of the study clearly articulated with a clear testable hypothesis stated?

-Is the study design appropriate to address the stated objectives?

-Is the population clearly described and appropriate for the hypothesis being tested?

-Is the sample size sufficient to ensure adequate power to address the hypothesis being tested?

-Were correct statistical analysis used to support conclusions?

-Are there concerns about ethical or regulatory requirements being met?

Reviewer #1: all: yes, except two missing information about statistics, as indicated below.

Reviewer #2: This is an interesting paper demonstrating the identification of a novel inhibitors of PIM kinase, which is found in stem cells of E. multilocularis metacestodes and is involved in cell cycle progression. The authors also demonstrate the interaction between PIM and CDC25. The paper is especially interesting due to the fact that it shows how inhibitors of human kinases can be further investigated through high-throughput in silico docking approaches, ending up with a novel inhibitor that has increased specificity for the parasite kinase over the human kinase. 

The study has been carried out with great care and expertise. Overall the paper shows good science, as expected from this group, and is very well written.

Reviewer #3: Yes, the manuscript met all the criteria mentioned for methods

**Results**

-Does the analysis presented match the analysis plan?

-Are the results clearly and completely presented?

-Are the figures (Tables, Images) of sufficient quality for clarity?

Reviewer #1: all: yes.

Reviewer #2: Overall this is a highly interesting contribution to the field. The results are clearly presented, and this includes the background of supplementary files. Figures and Tables are clearly described. 

The major backdrop of this paper is the lack of any in vivo data, which would confirm that the hopes that are being raised by the discovery that this group has made are actually justified. 

The drug discovery field is full of studies that show promising results in vitro, which are then often not translated into an animal model. More concretely for this study, we don't even know whether application of the novel compound to mice would kill these animals or not. If there is some background information available on that, the authors should show it.

Reviewer #3: Yes, the manuscript met all the criteria mentioned for results. Minor changes are suggested for Figures

**Conclusions**

-Are the conclusions supported by the data presented?

-Are the limitations of analysis clearly described?

-Do the authors discuss how these data can be helpful to advance our understanding of the topic under study?

-Is public health relevance addressed?

Reviewer #1: all: yes.

Reviewer #2: The major backdrop of this paper is the lack of any in vivo data, which would confirm that the hopes that are being raised by the discovery that this group has made are actually justified. 

The drug discovery field is full of studies that show promising results in vitro, which are then often not translated into an animal model. There are a thousand reasons why a compound will not work in an animal. More concretely for this study, we don't even know whether application of the novel compound to mice would kill these animals or not. If there is some background information available on that, the authors should show it. One of these compounds has been in phase 1 studies, and I am sure there is data providing information on PK properties of this drug, just to see whether the concentrations used in this paper are actually realistic. I believe such data should be discussed.

Inclusion of a small in vivo efficacy study, even if it turn out negative (e.g. the drugs has no or only limited effect in a mouse model), would give the reader a better understanding of the importance of the message that is conveyed here, and would surely make this study much more interesting.

Reviewer #3: Yes, the manuscript met all the criteria mentioned for conclusions.

**Editorial and Data Presentation Modifications?**

Reviewer #1: Only minor improvements required

Reviewer #3: Line 91: I prefer the word “niche” however the word department seems to be right.

Line 148: a dot is missing in “(GH09) [30]The”

Line 166: the abbreviature for days was already introduced in line 154. Please replace days by d

Paragraph starting at line 176: In my opinion, the authors take for granted that at higher luminescence more viability but, How do you know this?

Line 281: WormBase ParaSite, with capital letter in b.

Line 283: Domain instead of Dmain.

Line 313: replace seeSAR with SeeSAR.

Figure 1: for the sake of clarity, the pim kinase sould be remarked.

Figure 1: what do the percentages below the heatmap means? percentage of total viability? please, clarify.

 Figure 3: I like the pictures but I wonder if some kind of small drawing could be included showing schematically in which slice/position of the MV the pictures were taken

S1 Figure: the structure for the compound Z991902128 can´t be seen (please see file 17).

S2 Figure: for the sake of clarity, perhaps “expression levels of empim RNA in..” sounds clearer.

**Summary and General Comments**

Reviewer #1: The paper of Koike et al. deals with a PIM kinase of E. multilocularis as potential target for chemotherapy. In this comprehensive study, the authors provide in situ hybridization data showing stem cell activity of EmPIM. Y2H analyses revealed its interaction with E. multilocularis CDC25, a cell cycle regulator. Compounds known to inhibit human PIM caused deleterious effects on metacestode vesicles in vitro by preventing further development. Furthermore, the authors performed a high-throughput in silico modelling approach and identified a compound called Z196138710. This substance also affected cultured metacestode vesicles in vitro but showed less toxicity towards human HEK293T and HepG2 cells. The authors conclude that EmPIM is a promising target, and their study led to the identification of a novel small molecule compound with high effectivity against EmPIM and, thus, E. multilocularis.

The manuscript is very well written, and its content perfectly fits to the aims and scope of PLoS NTD. As such, it will be of high interest for the community. I have only little to suggest for improvement. 

Minor comments

line 96: define PK as abbreviation for protein kinase, and use it from this point on.

line 615: Fluence not in italics (as elsewhere in the text; see also lines 295-307)

Fig. 3, B: mentioned is “statistical analysis” but there is no statistic method mentioned, nor statistic values or indicators in the figure. 

Fig. 4/S1 Fig.: Is it really necessary to go with both figures? It seems there is enough overlap to fuse contents to one figure.

S2 Fig.: is this an n=1 experiment? If yes, indicate. If not, any statistics available?

References: harmonize the mixture of capital and small letter writing for paper titles according to the instructions for authors (see e.g. reference 48 vs others)

Reviewer #2: I have few points that the authors might want to consider in their revision:

Abstract and introduction

The authors indicate that it is necessary to identify novel compounds for the treatment of AE. This is of course undisputed, but fact is that benzimdazoles, after being applied for many decades, are made looking worse than they actually are. To state that they do not act parasiticidal is an oversimplification of the situation. This has been shown in a study carried out many years ago on 34 patients undergoing long-term albendazole treatment (2–25 years), which showed that after treatment stop 11 of 34 patients exhibited no recurrence of disease within 16–82 months, as determined by PT, CT scanning and serology, with immunocompetence being an important criterion for parasiticidal effects (Ammann et al, PLoS Negl. Trop. Dis. 9, e0003964), while two-thirds of AE cases were not cured. Clearly, liver damage due to extended treatment is a major concern, which renders this study an important one.

More specific comments:

Lane 320 ff: the initialscreen was done at one concentration only. Normally screenings are performed with different concentrations to study dose-responses, and then it would be possible to get an EC50 and a MIC, which would be much more informative. Maybe this is methodologically not possible in this system? The authors could give a short explanation. 

Lane 552: The results on metacestode integrity are clearly demonstrated. However, it is not clear to the reader what type of mature vesicles were assessed. How long have these vesicles been in culture, are they derived from stem cell cultures, did they already have protoscoleces, and is there a difference between older and younger vesicles? I can see that the methodology was referenced, and I am sure the authors have this information. Also, the assessment is based on visualization, which is good enough. When doing so, however, it is advised that this is done a blinded way (e.g. the one doing the assessment did not know what he/she is looking at). Any information on that should be given. Did the authors also consider to use another read-out found in the literature such as PGI or alkaline phosphatase assay? 

Incubation for 28 days: I might have missed that, but were there medium changes and at what interval? Is it known how long these drugs are stable under culture conditions? Please keep in mind that it is very unlikely that drug levels remain stable over longer periods in vivo. 

Lane 590: clearly, these inhibitors exert detrimental effects on metacestodes. However, to what extent is this relevant? It would be interesting to get a bit more information on the PK properties of these compounds. As one has undergone clinical trials at least with one of these compounds, there should be information on whether these concentrations showing an effect against E. multilocularis will ever be achieved in vivo. The concentration range that is active here appears rather high, and one then wonders whether the effects we see here are not based on off-target effects (even not necessarily based on inhibition of another kinase) that might occur. I am not questioning the results per se, but this aspect could be discussed.

Lane 602: is the cause for the interruption of the heart rhythm (long QT syndrome) based on a specific interaction of SGI-776 with the the human Ether-à-go-go-Related Gene (hERG)? This is a common stumbling step in drug development. Clearly, the in silico approach appears to be a novel and highly versatile tool to obtain novel molecules such as Z196138710 with profound activities and in this case more specificity, and the results obtain are encouraging in terms of efficacy and specificity, but I would advise the authors to look into hERG inhibition of their novel compound early on. 

Lane 656: Screening of mammalian cells employed HepG2 and HEK293T cells, which are immortalized cell lines. It would be advisable to also assess cultures of non-immortalized cells or primary cell cultures of different origin.

Reviewer #3: This is an interesting manuscript in which the authors identified the proto-oncogene EmPIM kinase as a promising target for anti-AE chemotherapy. The apparently well performed in situ hybridization assays indicated its expression in parasite stem cells. By yeast two-hybrid assays, the authors showed interaction of EmPIM with E. multilocularis CDC25, suggesting an involvement of EmPIM in parasite cell cycle regulation. 

To improve compound specificity for EmPIM, the authors applied a high throughput in silico modelling approach, leading to the identification of compound Z196138710. Which, when applied to in vitro cultured metacestode vesicles and parasite cell cultures, proved equally detrimental as SGI-1776 and CX-6258, but displayed significantly reduced toxicity towards human HEK293T and HepG2 cells.

Overall, the work is interesting and relevant as a potential alternative for future anti-AE chemotherapy.

However, I have some doubts that in my opinion, merit some consideration/reflection.

Major:

To the light of the results presented here, the compound Z196138710 seems to be promissory in echinococcosis therapy.

However, I wonder if a more complete curve (with more than 3 points) could be performed for this compound in terms of viability, intact vesicles or regenerated vesicles. I´m not suggesting to do 3 curves, but only one which could allow to get an IC50 value or a clearer picture about the nature of the inhibition. 

I wonder if any information exists about the plasma levels of kinase inhibitors in mammals. If the answer to the previous question is affirmative and based on the dose response curve obtained for Z196138710, Could the plasma levels of kinase inhibitors be in the range necessary for therapeutic intervention of Echinococcosis? 

Have the authors attempted to perform silencing of PIM in PC to evaluate the relevance of this kinase in cell viability and MV formation?

PLOS authors have the option to publish the peer review history of their article (what does this mean?). If published, this will include your full peer review and any attached files.

Reviewer #1: No

Reviewer #2: No

Reviewer #3: No

Figure Files:

Data Requirements:

Reproducibility:

References

---

## [Decision Letter · Decision Letter 1]

20 Sep 2022

Dear Prof. Brehm,

We are pleased to inform you that your manuscript 'Targeting Echinococcus multilocularis PIM kinase for improving anti-parasitic chemotherapy' has been provisionally accepted for publication in PLOS Neglected Tropical Diseases.

Best regards,

Bruce A. Rosa

Academic Editor

Makedonka Mitreva

Section Editor

The reviewers who have agreed to review the revised manuscript have agreed that the authors have made substantial improvements and that the manuscript is ready for publication.

Reviewer's Responses to Questions

**Key Review Criteria Required for Acceptance?**

**Methods**

-Are the objectives of the study clearly articulated with a clear testable hypothesis stated?

-Is the study design appropriate to address the stated objectives?

-Is the population clearly described and appropriate for the hypothesis being tested?

-Is the sample size sufficient to ensure adequate power to address the hypothesis being tested?

-Were correct statistical analysis used to support conclusions?

-Are there concerns about ethical or regulatory requirements being met?

Reviewer #1: all yes

Reviewer #2: No comments on this section, the authors have done a great job in revising this paper

**Results**

-Does the analysis presented match the analysis plan?

-Are the results clearly and completely presented?

-Are the figures (Tables, Images) of sufficient quality for clarity?

Reviewer #1: all yes

Reviewer #2: The results are clearly presented, and all figures and tables and images are clearly presented

**Conclusions**

-Are the conclusions supported by the data presented?

-Are the limitations of analysis clearly described?

-Do the authors discuss how these data can be helpful to advance our understanding of the topic under study?

-Is public health relevance addressed?

Reviewer #1: all yes

Reviewer #2: All points of criticism were addressed by the authors and were either changed or they provided convincing arguments not to change their statements. All relevant points are addressed.

**Editorial and Data Presentation Modifications?**

Reviewer #1: (No Response)

Reviewer #2: No modifications required

**Summary and General Comments**

Reviewer #1: The authors made a fine effort to improve their interesting manuscript, which from my perspective can be accepted for publication.

Reviewer #2: The authors have done a great job in revising their manuscript. Congratulations!

PLOS authors have the option to publish the peer review history of their article (what does this mean?). If published, this will include your full peer review and any attached files.

Reviewer #1: No

Reviewer #2: **Yes: **Andrew Hemphill

---

## [Editor Report · Acceptance letter]

28 Sep 2022

Dear Prof. Brehm,

We are delighted to inform you that your manuscript, "Targeting Echinococcus multilocularis PIM kinase for improving anti-parasitic chemotherapy," has been formally accepted for publication in PLOS Neglected Tropical Diseases.

Best regards,

Shaden Kamhawi

co-Editor-in-Chief

Paul Brindley

co-Editor-in-Chief
